# Evaluating the Sensitivity of Heat Wave Definitions among North Carolina Physiographic Regions

**DOI:** 10.3390/ijerph191610108

**Published:** 2022-08-16

**Authors:** Jagadeesh Puvvula, Azar M. Abadi, Kathryn C. Conlon, Jared J. Rennie, Hunter Jones, Jesse E. Bell

**Affiliations:** 1Department of Environmental, Agricultural and Occupational Health, College of Public Health, University of Nebraska Medical Center, Omaha, NE 68198, USA; 2Department of Public Health Sciences, University of California Davis, One Shields Ave, Davis, CA 95616, USA; 3National Centers for Environmental Information, Asheville, NC 28801, USA; 4Medical Sciences Interdepartmental Area, Office of Graduate Studies, University of Nebraska Medical Center, Omaha, NE 68198, USA; 5School of Natural Resources, University of Nebraska-Lincoln, Lincoln, NE 68583, USA; 6Daugherty Water for Food Global Institute, University of Nebraska, Lincoln, NE 68583, USA

**Keywords:** heat wave, heat-related illness, early heat−health warning systems

## Abstract

Exposure to extreme heat is a known risk factor that is associated with increased heat-related illness (HRI) outcomes. The relevance of heat wave definitions (HWDs) could change across health conditions and geographies due to the heterogenous climate profile. This study compared the sensitivity of 28 HWDs associated with HRI emergency department visits over five summer seasons (2011–2016), stratified by two physiographic regions (Coastal and Piedmont) in North Carolina. The HRI rate ratios associated with heat waves were estimated using the generalized linear regression framework assuming a negative binomial distribution. We compared the Akaike Information Criterion (AIC) values across the HWDs to identify an optimal HWD. In the Coastal region, HWDs based on daily maximum temperature with a threshold > 90th percentile for two or more consecutive days had the optimal model fit. In the Piedmont region, HWD based on the daily minimum temperature with a threshold value > 90th percentile for two or more consecutive days was optimal. The HWDs with optimal model performance included in this study captured moderate and frequent heat episodes compared to the National Weather Service (NWS) heat products. This study compared the HRI morbidity risk associated with epidemiologic-based HWDs and with NWS heat products. Our findings could be used for public health education and suggest recalibrating NWS heat products.

## 1. Introduction

A heat wave is often described as an acute episode of one or more consecutive days with temperatures or heat indices exceeding a threshold value [1]. However, no standard definition exists to identify heat waves [2]. Heat waves are typically classified using a synoptic (e.g., air mass, temperature-humidity index), physiologic (e.g., Environmental Stress Index, Wet Bulb Global Temperature), or epidemiologic approach [3]. Hajat et al. (2010) reported that epidemiologic-based algorithms (temperature−mortality relationship) identified the days with higher heat-related mortality.

Heat waves are associated with an increased risk of HRI outcomes [4]. In the United States (US), roughly 700 heat-related deaths per year are attributable to ambient temperature exposure [5]. The frequency and intensity of heat waves have been on the rise since the industrial revolution and are likely to increase in the future due to climate change [6,7]. In Philadelphia, heat-related risk communication, along with the NWS warnings, played a crucial role in minimizing up to three heat-related deaths per day that are associated with extreme heat exposure [8,9]. In the US, public health departments rely on heat products (e.g., excessive heat watch, heat advisory, and excessive heat warning) from the local National Weather Service Weather Forecast Office (NWS-WFO) to communicate heat−health risks. According to Weinberger et al. (2018), the NWS heat products moderately reduced the impact of extreme heat on human health, but the human health risk associated with ambient temperature is not a resolved issue [10].

Despite the early heat warnings, the average annual percentage of HRI emergency department visits in North Carolina has increased by 19% over the past decade [11,12]. The rise in HRI risk could be due to population vulnerabilities (age, economic status, and occupation) and higher thresholds for temperature and humidity set by NWS for issuing heat alerts [13]. Multiple researchers attempted to characterize human health risks associated with HWDs [3,14,15,16,17]. Epidemiologic studies evaluating the association between outdoor temperature exposure (e.g., heat waves or ambient temperature) and human health are generally focused on extreme events, where the human health outcome is typically measured as cause-specific or all-cause mortality [17,18]. A meta-analysis on the HWD evaluation studies summarized that the researchers included in this review generalized the warnings to a larger geographic area, such as a state or a group of states [17]. Generalizing heat warning systems over larger geography may not be ideal due to heterogeneity in exposures, population vulnerability, and exposure−outcome associations [19]. Physiographic or sub-regional-scale heat warning systems that account for meteorological heterogeneity and are specific to health conditions were found to play a role in minimizing the human health risks associated with heat waves [8,16,20,21]. 

In this study, we assessed the association between multiple HWDs and HRI emergencies in North Carolina physiographic regions. We then compared the statistical model performance across the HWDs included in this study and assessed their association with HRI emergency visits in North Carolina. Additionally, we aimed to compare the HWD with the best model performance from our study with the NWS extreme heat alerts.

## 2. Materials and Methods

This study is focused on five summer seasons (1 May–30 September) from 2011 to 2016 among the three physiographic regions (Coastal, Piedmont, and Mountain) in North Carolina. The year 2013 was excluded in this study due to data availability constraints. Additionally, the Mountain region was excluded from the analysis as 50.13% of the data were censored due to low HRI visits.

### 2.1. Data

#### 2.1.1. Heat Metrics

Daily mean, minimum, and maximum temperatures were obtained from the Global Historical Climate Network–Daily (GHCN-D) database [7,22]. Dew point data were obtained from the Parameter-elevation Regression on Independent Slopes Model (PRISM) database [23]. The station-based temperature measurements and gridded dew point data were aggregated by physiographic region. The daily maximum apparent temperature and relative humidity were estimated using daily maximum temperature and dew point using heat.index.function and dewpoint.to.humidity functions available from the weathermetrics package in R [24].

#### 2.1.2. Heat Wave Definitions

Twenty-eight HWDs (Table 1) associated with human health outcomes were adopted from the existing literature and were included in this study [14,15,25,26]. These HWDs were classified based on four factors: (1) a heat metric (daily mean, minimum, maximum, and apparent temperatures), (2) duration (number of days), (3) threshold type (relative/absolute), and (4) threshold intensity. Among the two threshold types, relative threshold-based definitions account for cumulative heat exposure (2+ or 3+ consecutive days), and definitions based on absolute threshold are based on single heat day exposure. The HWDs based on daily temperature as a metric and a relative threshold were classified using four percentile values (99, 98, 95, 90) as threshold intensity. The percentile threshold values were calculated using historical observations over the summer season from 1895 to 2016. The definitions based on the apparent temperature as metric and a relative threshold were classified using three percentile threshold values (95, 90, 85) as threshold intensity. 

Among the 28 HWDs (HW_01 to HW_28) included in this study, 24 are based on relative threshold values, and four HWDs are based on the absolute threshold value. Three of the 27 HWDs using relative threshold values were based on maximum apparent temperature as the heat metric. The remaining 24 HWDs were based on the daily mean (HW_01-HW_08), maximum (HW_09-HW_16), and minimum (HW_18-HW_25) temperature values as the heat metric. Additionally, one HWD using an absolute threshold value is based on daily maximum temperature as the heat metric (Table 1). Using the 28 HWDs, we categorized the summer days during the study period as a heat wave and non-heat wave days using a binary variable to indicate heat waves.

#### 2.1.3. National Weather Service—Heat Products

The heat products released by the NWS during the study period were retrieved from the Iowa Environmental Mesonet [27]. During the study period, the NWS heat products (heat advisories and excessive heat warnings) were released by the three NWS-WFOs (ILM—Wilmington, MHX—Newport/Morehead city, and RAH—Raleigh) located in North Carolina, which were included in this study. The WFOs ILM and MHX cover most of the Coastal region, and RAH covers the Piedmont region [28]. Among the three WFOs, heat products released by the ILM and MHX follow the NWS procedural directive. The RAH WFO is the only center in North Carolina that collaborated with health partners to revise the heat products [29]. The heat products from RAH are based on local conditions such as maximum temperature, sunlight, nighttime temperature, heterogeneity between rural and urban temperatures, and knowledge from historical weather conditions [29]. The heat products used in the three WFOs in North Carolina are based on the following criteria. A heat advisory is released during the days when the daytime heat index value is between 100 and 105 °F [30]. An excessive heat warning is released if the daytime heat index forecast value is between 105 and 110 °F [30].

We extracted the start and end dates and County information from the Iowa Environmental Mesonet heat product archives. The County information was aggregated to the North Carolina physiographic region scale to match the spatial resolution of the health data included in this study. We considered NWS heat wave days among physiographic regions if one or more counties by WFOs within the physiographic regions had heat warning/advisories (Appendix A). The NWS heat products were represented using a binary variable (NWS_HW) to identify the days with NWS alerts on a daily scale by physiographic region.

#### 2.1.4. Heat-Related Illness

Daily HRI-related emergency department visit data were obtained as an aggregate count per day per physiographic region from the North Carolina Disease Event Tracking and Epidemiologic Collection Tool (NC DETECT) surveillance program maintained by the North Carolina Division of Public Health (NC DPH) [31]. Heat-related illnesses were defined using ICD-9 CM codes with E992/E900.0/E900.0/E900; ICD-10 CM codes within T67/X30/X32; and various keywords from the chief complaint/triage notes [31,32]. The days with HRI emergency department visits fewer than five were censored, amounting to 28.81% (219) of observations from the Coastal and 28.94% (220) in the Piedmont region. The days with censored HRI emergency visits in the Coastal and Piedmont regions were imputed by the median value of 3 visits per day.

### 2.2. Statistical Analysis

The sensitivity of 28 HWDs were compared using the Akaike Information Criterion (AIC) value corresponding to the model fit [33,34] evaluating the HRI morbidity rate associated with heatwaves included in this study. AIC is a metric that is a balance between model accuracy and penalty due to complexity, commonly used to measure the optimal model fit (Equation (1)) [35]. A smaller AIC value (close to −∞) represents an optimal fit [36].
(1)AIC=goodness of fit+penalty

The HRI rate ratios corresponding to the 28 HWDs included in this study were estimated using the Generalized Linear Model (GLM) and assuming negative binomial distribution to account for outcome overdispersion. To compare the HRI risk across physiographic regions, the regression model was adjusted for population density by using the 2010 decennial population by region as an offset term [37]. To estimate the direct effect of HWDs, the statistical models using HWDs based on temperature as a heat metric were adjusted for relative humidity, and NWS heat wave alert days were added as covariates to adjust for potential confounding effects and effect modification. Similarly, the statistical models with HWDs using apparent temperature as a heat metric were adjusted for NWS heat alerts. Additionally, to account for temporal autocorrelation, we adjusted the statistical models mentioned above for the day of the week (weekday/weekend (binary)), month (factor), and year (factor) (Figure 1 and Equation (2)). In Equation (2), HW is a binary variable that represents HWDs, RH represents relative humidity, NWS-HW represents NWS heat wave alerts, and TS represents the time series variables (day of week, month, and year).
(2)$$\log\left(\frac{E\left(\mathrm{HRI}\;\mathrm{count}\right)}{\mathrm{population}}\right)=\beta_{0}+\beta_{\mathrm{HW}}+\beta_{\mathrm{RH}}+\beta_{\mathrm{NWS}\_\mathrm{HW}}+\beta_{\mathrm{TS}}+\varepsilon$$

We processed 28 statistical models stratified by physiographic region to obtain the rate ratio (RR) and 95% confidence intervals. The sensitivity of HWDs was evaluated by comparing the AIC values across HWDs by physiographic region that were generated from the GLM output. The statistical model with the lowest AIC value among the 28 HWDs was considered the optimal HWD (Coastal: HW_15 and Piedmont: HW_07). We then compared the overlap between the days considered as heat waves from this study and the NWS heat products using the Chi-Square test [38]. The analysis was conducted using R version 4.0.3 and MASS package version 7.3 [39]. 

## 3. Results

### 3.1. Heat Wave Definition—Sensitivity

Among the 28-heat metrics included in this study, the HWD using maximum temperature had the best fit with HRI morbidity in the Coastal region and mean temperature for the Piedmont region. The HWDs based on a moderate (90th) percentile threshold for Coastal and Piedmont regions had a more optimal model fit than the HWDs based on extreme threshold (99th, 98th, and 95th percentile) values.

In the Coastal region, the HWD based on daily maximum temperature as a heat metric with a threshold value >90th percentile for two or more consecutive days (HW_15) had the optimal model fit (lowest AIC value) to estimate the HRI morbidity compared to the HWDs included in this study. We did not observe a similar result for the Piedmont region. In the Piedmont region, the HWD based on daily mean temperature as a heat metric with a threshold value >90th percentile for two or more consecutive days (HW_07) had the optimal model fit to estimate the HRI morbidity. In the Coastal region, the HWD HW_15 is associated with a 2.75 (95% CI 2.40–3.08) times higher HRI morbidity rate during heat wave days than the non-heat wave days. In the Piedmont region, the HWD HW_07 is associated with a 2.72 (95% CI 2.46–3.01) times higher HRI morbidity rate compared to the non-heat wave days (Figure 2).

Using the HW_15 definition, 27% (190/704) of the days in the Coastal and using the HW_07 HWD, 34% (241/719) of the days in the Piedmont region were flagged as heat wave days during the study period (Table 2). There are an average of six HRI ED visits per day in the Coastal region during the heat wave days based on HW_15 and an average of eight HRI ED visits per day during the heat wave days based on HW_07 in the Piedmont region (Table 2). The frequency of heat wave days in the Piedmont region using the HW_07 definition was 24% higher than in the Coastal region using the HW_15 definition. About 72% of the heat wave days from the Coastal region matched with the Piedmont region. During the study period, we observed a lower number of heat wave days during the summer of 2014. The frequency of heat wave days was higher in July than in other summer months, based on the epidemiologic relationship based HWD (Coastal: HW_15; Piedmont: HW_07).

### 3.2. Comparing Epidemiologic-Based Heat Wave Definition and NWS Heat Products

During the study period, NWS flagged 26 days in the Coastal and 18 days in the Piedmont regions as heat waves. The NWS heat wave days overlapped with the optimal HWD identified in this study (HW_15 for Coastal and HW_07 for Piedmont). In the Coastal region, there was a significantly higher (six-times) number of heat wave days based on HW_15 than the NWS heat alerts (McNemar χ^2^ = 158.15, df = 1; *p* < 0.05). Similarly, the Piedmont region had a significantly higher (13 times) number of heat waves based on HW_07 than the NWS heat alerts (McNemar χ^2^ = 219.04, df = 1, *p* < 0.05) (Figure 3).

## 4. Discussion

This study compared the optimal model fit using AIC values across the 28 HWDs. Additionally, it compared the heat wave days flagged using the optimal definition identified from this study to the NWS heat products. We observed that the HWD based on the maximum temperature (HW_15) had an optimal performance for the Coastal region and the mean temperature based HWD (HW_07) for the Piedmont region. The HWDs mentioned above were associated with a 2.75 (95% CI 2.40–3.08) times higher rate of HRI morbidity in the Coastal region and a 2.72 (95% CI 2.46–3.01) times higher HRI morbidity rate in Piedmont than the non-heat wave days. During the study period, our results suggest an excess of 33 heat wave days per summer season in the Coastal and 45 in the Piedmont region based on the HWDs HW_15 for Coastal and HW_07 for Piedmont. During the summer, most days in July were flagged as vulnerable to heat-related emergencies while using the HWDs from this study.

Heterogeneity while evaluating the HWDs across the US climate regions or sub-regions was well established in the literature [15,16,21]. To address the meteorological heterogeneity within North Carolina, we evaluated the heat waves stratified by physiographic delineations by clustering the administrative boundaries into physiographic regions. In North Carolina, there are two Weather Forecast Offices (WFOs) covering the Coastal region and a WFO covering the Piedmont region. The WFO covering the Piedmont region collaborated with the regional health partners to revise the heat thresholds. The WFO that overlaps the Piedmont region considered revising its heat threshold based on the local conditions for optimization [29]. Independent WFOs setting heat product thresholds at a sub-regional scale would be beneficial for the climate-related heterogeneity across the administrative boundaries. However, a study discussing the heat products across the US reported that the three WFOs across the Coastal and Piedmont regions follow the same threshold value and criteria to release heat products [29]. In contrast to the homogenous heat product thresholds across the physiographic regions, our results suggest heterogeneity of the HWDs between the Coastal and Piedmont regions. 

The results from our study overlapped with the observations from a previous study that compared the sensitivity of HWDs across San Diego climate zones, using heat-related hospitalizations as an outcome [21]. McElroy et al. (2020) reported using daily maximum temperature above the 90th percentile (29.11 °C) in a day as a criterion for HWDs in the Coastal region to be most efficient, using heat-related hospitalizations as an outcome. In this study, we observed that the HWD based on daily maximum temperatures above the 90th percentile (31.97 °C) for two or more days had an optimal fit with HRI morbidity in the Coastal region. Additionally, we observed that the HWD based on daily mean temperature was optimal for the Piedmont region. In contrast, McElroy et al. (2020) reported that HWDs based on daily maximum and the minimum temperature had the most impact on the Inland and Desert regions of San Diego. The major difference between our findings and McElroy et al. (2020) is focused on the duration criterion for defining heat waves. We observed that the HWDs using two or more days as a duration criterion had an optimal fit, compared to McElroy et al. (2020), who reported that absolute thresholds were most efficient.

Early heat−health warning systems play a crucial role in systematically minimizing the risks associated with outdoor temperature exposure [8]. Multiple studies attempted to characterize a gold standard HWD, where most of these studies compared the sensitivity of the HWDs in the context of mortality [14,15,17]. Vaidyanathan et al. (2016) evaluated the sensitivity of several HWDs and their association with heat-related deaths by comparing the effect estimates (extreme heat effect). Similarly, Anderson and Bell (2009) evaluated the sensitivity of HWDs based on the percent increase in relative risk associated with heat-related mortality. McElroy et al. (2020) evaluated heat waves by climate zones, using the attributable risk associated with heat-related hospitalizations. These studies assessed the optimal HWD by comparing the effect estimates/relative or attributable risks associated with health outcomes. We compared and identified an optimal HWD per North Carolina physiographic regions using the model fit metric of the lowest AIC value (model fit) instead of the effect estimates (strength of association).

Using AIC as a metric, we compared 28 HWDs and identified an HWD with an optimal model fit. The HWD (HW_15) using daily maximum temperature with a > 90th percentile value for two or more consecutive days had an optimal fit for the Coastal region. Similarly, using daily mean temperature with a threshold > 90th percentile value for two or more consecutive days was optimal for the Piedmont region compared to the HWDs included in this study. During the study period, 27% of the summer days in the Coastal and 34% in the Piedmont region were flagged as vulnerable to HRI emergencies by our definition. In contrast, the NWS released heat products during ~2.5% of the summer days. During the study period, there were an average of 783 (6 per day) HRI emergency visits per summer season during the days flagged as vulnerable based on the HW_15 definition in the Coastal region and 1152 (8 per day) HRI emergencies per summer season in the Piedmont region during the days flagged as vulnerable to heat-related emergencies using the HW_07 definition. Abasilim and Friedman (2021) reported about 16 heat-related hospitalizations per day during the summer days without NWS excess heat warnings in Illinois [40].

Our results could be influenced by the interaction between vulnerability factors and risk perception. Additionally, our results are subjective to a variety of unmeasured biases driven by human vulnerabilities such as co-existing medical conditions, occupational vulnerabilities, demographics (age, gender, race, education, urbanicity), and socioeconomic factors (wealth, employment, housing) that were identified to exacerbate the risk of heat-related illnesses [41,42,43,44,45,46,47]. Additionally, our results could be influenced by effect modifiers such as human behavioral factors that include knowledge on heat risk sensitivity, external locus of control, and emotional and cognitive factors that heavily alter the risk perception of heat warnings [48,49,50]. Further studies evaluating HWDs using mixed methods by considering quantitative information from human vulnerability characteristics and qualitative information from heat risk perception could strengthen the heat−health risk ascertainment.

## 5. Conclusions

Our results showed heterogeneity of the optimal HWDs among the Coastal and Piedmont regions in North Carolina. Additionally, the threshold values associated with the optimal HWDs were smaller compared to the NWS thresholds for the North Carolina physiographic regions. Our results suggest recalibrating the HWDs used by the NWS WFOs in North Carolina.

## Figures and Tables

**Figure 1 ijerph-19-10108-f001:**
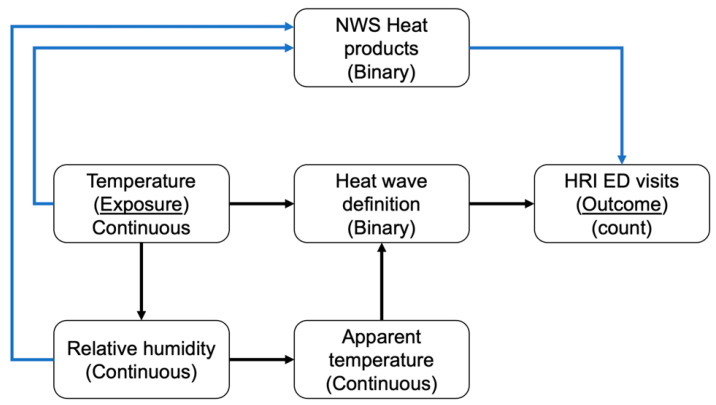
Conceptualization of evaluating the direct effect of temperature or apparent temperature on HRI ED visits. We assumed that the association between temperature and HRI is mediated through NWS heat products. Additionally, relative humidity is influenced by temperature. To evaluate the association between HWDs based on temperature and HRI, we adjusted for relative humidity and NWS heat products. We adjusted for NWS heat products while evaluating the association between HWDs based on apparent temperature and HRI ED visits.

**Figure 2 ijerph-19-10108-f002:**
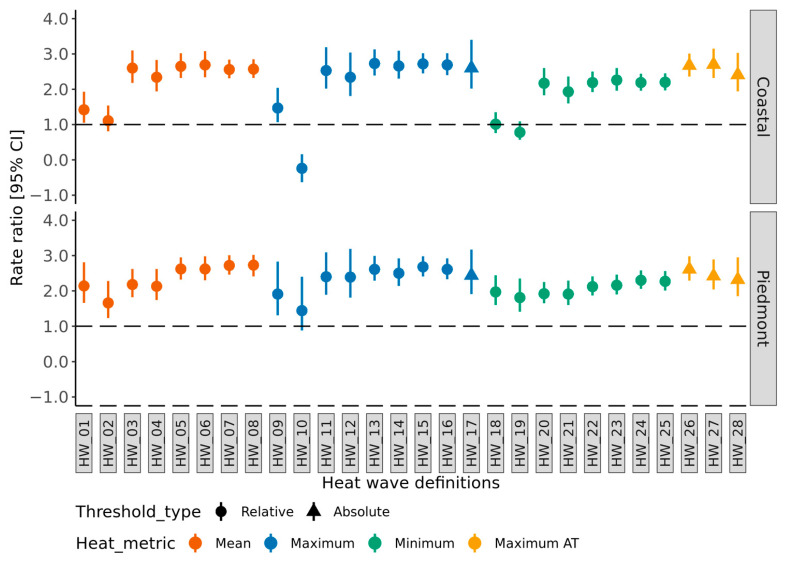
HRI rate ratio corresponding to heat wave definitions. The rate ratios and the corresponding 95% confidence intervals were generated using the generalized linear model (GLM), assuming a negative binomial distribution. The *X*-axis represents distinct heat wave definitions, stratified by North Carolina physiographic regions and grouped by metric and threshold type. The *Y*-axis represents the HRI morbidity rate ratio, which could be interpreted as an increase/decrease in HRI morbidity rate during a heat wave day compared to a non-heat wave day.

**Figure 3 ijerph-19-10108-f003:**
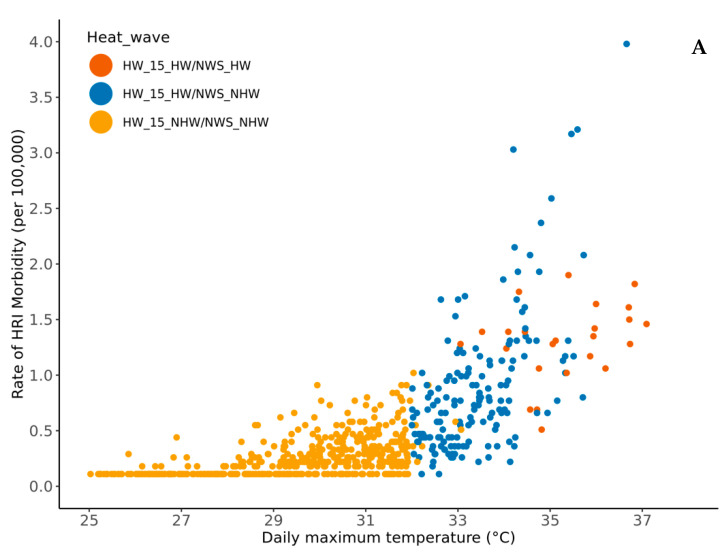
Comparison of the effective heat wave definition with the NWS-Heat alerts. The *X*-axis represents daily temperature in degrees Celsius (Coastal: maximum temperature and Piedmont: minimum temperature). The *Y*-axis represents the daily rate of heat-related illness morbidity per 100,000 population members. Each dot represents an observation corresponding to daily temperature and the rate of HRI morbidity during the study period. Panel (**A**) represents the Coastal region and panel (**B**) represents the Piedmont region. The dots in the scatter plot are color-coded with three possible combinations: (1) Red: Categorized as heat wave day from our results (HW_15 or HW_07) and the NWS; (2) Blue: Categorized as heat wave day only based on our result; (3) Beige: Not flagged as a heat wave day from our results nor the NWS.

**Table 1 ijerph-19-10108-t001:** Description of the heat wave definitions.

Definition	Heat Metric	Duration (No. of Days)	Threshold Type	Threshold Intensity	Coastal			Piedmont		
					Threshold (°C)	HW Days	AIC	Threshold (°C)	HW Days	AIC
HW_01	Mean temperature	2 + consecutive	Relative	>99th percentile	29.16	33	4475.7	28.40	29	4817.9
HW_02		3 + consecutive	Relative	>99th percentile		25	4480.7		21	4843.0
HW_03		2 + consecutive	Relative	>98th percentile	28.57	58	4362.9	27.66	61	4778.4
HW_04		3 + consecutive	Relative	>98th percentile		46	4396.7		49	4796.9
HW_05		2 + consecutive	Relative	>95th percentile	27.65	105	4287.3	26.70	136	4629.7
HW_06		3 + consecutive	Relative	>95th percentile		97	4293.5		118	4656.9
HW_07		2 + consecutive	Relative	>90th percentile	26.65	217	4216.2	25.63	241	4547.9
HW_08		3 + consecutive	Relative	>90th percentile		195	4211.8		227	4551.8
HW_09	Maximum temperature	2 + consecutive	Relative	>99th percentile	35.04	22	4475.4	35.52	20	4842.6
HW_10		3 + consecutive	Relative	>99th percentile		16	4479.7		14	4852.4
HW_11		2 + consecutive	Relative	>98th percentile	34.31	38	4413.7	34.69	38	4799.8
HW_12		3 + consecutive	Relative	>98th percentile		30	4435.3		32	4813.2
HW_13		2 + consecutive	Relative	>95th percentile	33.13	98	4283.5	33.30	109	4665.9
HW_14		3 + consecutive	Relative	>95th percentile		80	4318.0		85	4716.0
HW_15		2 + consecutive	Relative	>90th percentile	31.97	190	4192.6	31.94	194	4583.4
HW_16		3 + consecutive	Relative	>90th percentile		168	4234.4		180	4615.9
HW_17		1-day	Absolute	>35 °C	35.00	26	4423.1	35.00	34	4801
HW_18	Minimum temperature	2 + consecutive	Relative	>99th percentile	23.86	29	4488.1	21.86	45	4810.3
HW_19		3 + consecutive	Relative	>99th percentile		25	4479.1		31	4831.9
HW_20		2 + consecutive	Relative	>98th percentile	23.36	56	4401.1	21.40	83	4783.6
HW_21		3 + consecutive	Relative	>98th percentile		46	4433.5		65	4799.5
HW_22		2 + consecutive	Relative	>95th percentile	22.57	108	4352.0	20.63	156	4722.0
HW_23		3 + consecutive	Relative	>95th percentile		94	4353.0		136	4722.6
HW_24		2 + consecutive	Relative	>90th percentile	21.64	223	4305.5	19.72	265	4670.3
HW_25		3 + consecutive	Relative	>90th percentile		199	4303.0		235	4693.7
HW_26	Maximum apparent temperature	1-day	Absolute	>95th percentile	37.21	36	4254.0	35.26	27	4659.0
HW_27		1-day	Absolute	>90th percentile	36.20	71	4319.9	35.92	58	4749.8
HW_28		1-day	Absolute	>85th	35.47	106	4415.7	36.95	98	4799.1

HW (heat wave) days per Coastal and Piedmont region represent the cumulative number of days during the study period that are categorized as heat wave days corresponding to the heat wave definitions.

**Table 2 ijerph-19-10108-t002:** Frequency of heat wave days and HRI ED visits in North Carolina physiographic regions.

		Month									
		May		June		July		August		September	
		a	b	a	b	a	b	a	b	a	b
**2011**	HW	3	1	21	14	22	25	10	15	0	0
	ED	65	37	309	334	225	417	195	302	0	0
**2012**	HW	0	0	5	6	23	27	0	8	2	6
	ED	0	0	45	95	306	520	0	112	35	52
**2014**	HW	0	0	6	7	7	11	0	5	5	6
	ED	0	0	101	141	103	149	0	55	68	92
**2015**	HW	0	0	15	16	14	19	5	9	2	3
	ED	0	0	817	897	306	569	117	177	25	54
**2016**	HW	0	0	5	8	24	26	19	24	2	5
	ED	0	0	80	204	721	950	366	499	30	106

a—Coastal; b—Piedmont; HW—number of heat wave days using the definition from this study (Coastal: HW_15; Piedmont: HW_07) and excluding the days that overlapped with the heat wave days flagged by the NWS; ED—number of HRI emergency department visits corresponding to the heat wave days.

## Data Availability

Temperature data are available the Global Historical Climatology Network daily (GHCNd) database, [Dataset]. Available at: https://www.ncei.noaa.gov/products/land-based-station/global-historical-climatology-network-daily (accessed on 12 April 2019); Precipitation and dew point data is available at the PRISM Climate Group database, [Dataset]. Available at: https://www.prism.oregonstate.edu/recent/ (accessed on 3 September 2019); Heat-related emergency department visit data could be requested from the North Carolina Department of Health and Human Services, Available at: https://ncdetect.org/data/ (accessed on 26 February 2019).

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
