# Peer review of "Evaluating the Sensitivity of Heat Wave Definitions among North Carolina Physiographic Regions"

_ijerph, 2022, doi:10.3390/ijerph191610108_

Round 1
Reviewer 1 Report
Excellent work!
Just a very few commentaries.
Additionally, we observed that the optimal heat wave definitions from this study captured moderate and frequent heat episodes than the national weather service (NWS) heat products ...
the sentence seems unclear to me.
However, there is no standard definition to identify heat waves [2].
However, no standard definition exists to identify heat waves [2].
This study assessed...
The objective of this study is to...
This study is focused on ...
Consider rewriting these sentences.
I counted 50 times "heat wave definitions",
perhaps use an abbreviation (e.g. HWD), it would make the readings faster.
A meta-analysis on the heat wave definition evaluation studies summarized that the studies researches included in this review generalized the warnings to a larger geographic area, such as a state or a group of states [17].
Author Response
Reviewer 1:
Additionally, we observed that the optimal heat wave definitions from this study captured moderate and frequent heat episodes than the national weather service (NWS) heat products ...
the sentence seems unclear to me.
Thank you, for your feedback. We revised line 27-29, “The heat wave definitions with optimal model performance included in this study captured moderate and frequent heat episodes compared to the National Weather Service (NWS) heat products.”
However, there is no standard definition to identify heat waves [2].
However, no standard definition exists to identify heat waves [2].
Thank you, for your feedback. We revised line 37 “However, no standard definition exists to identify heat waves.”
This study assessed...
The objective of this study is to...
This study is focused on ...
Consider rewriting these sentences.
Thank you, for your comments. We revied lines 73-76, “In this study, we assessed the association between multiple heat wave definitions and HRI emergencies in North Carolina physiographic regions. We then compared the statistical model performance across the heat wave definitions included in this study and assess their association with HRI emergency visits in North Carolina.”
I counted 50 times "heat wave definitions",
perhaps use an abbreviation (e.g. HWD), it would make the readings faster.
Thank you, for your comment. We abbreviated heat wave definition with HWD throughout the manuscript.
A meta-analysis on the heat wave definition evaluation studies summarized that the studiesresearches included in this review generalized the warnings to a larger geographic area, such as a state or a group of states [17].
Thank you, for your feedback. We revised line 64-67, “A meta-analysis on the heat wave definition evaluation studies summarized that the researchers included in this review generalized the warnings to a larger geographic area, such as a state or a group of states [17].”
Reviewer 2 Report
This paper mainly temperature response of heat-related illness (HRI) from the 28 heatwave definitions. Although this study will provide many readers to consider heatwave analyses, a problem remains for acceptance: (1) application of AIC and (2) heat stress index.
(1) Application of AIC: Authors seems to compare AIC values for different data structures of heatwave events. For the same structure of data set, the AIC should be generally used to explain combinations of explanatory variables as the best model. I’m not sure if it was appropriate to use AIC instead of RMSE, MAE, and R^2.
(2) Heat stress index: Why does not authors use heat stress index of WBGT or Humidex as an explanatory variable? The best fit model of HRI may be detected by WBGT with lower AIC rather than a single meteorological element.
Author Response
Application of AIC: Authors seems to compare AIC values for different data structures of heatwave events. For the same structure of data set, the AIC should be generally used to explain combinations of explanatory variables as the best model. I’m not sure if it was appropriate to use AIC instead of RMSE, MAE, and R^2.
Thank you for your comments. Yes, you are correct, we could have used either AIC, RMSE, MAE or R-squared values to compare the model performance. In this study we have used AIC values to compare the model fit as the AIC metric is a combination of goodness of fit that compares the fit between observed data to the predicted values along with penalty for number of metrics to reduce model complexity. Whereas the other model evaluation metrics such as RMSE, MAE, and r-squared purely focus on comparing the fit between observed vs predicted values. Our rationale to use AIC is to minimize bias due to over-adjusting for variables.
(2) Heat stress index: Why does not authors use heat stress index of WBGT or Humidex as an explanatory variable? The best fit model of HRI may be detected by WBGT with lower AIC rather than a single meteorological element.
Thank you for your comment. In this study we used three definitions (HW_26, HW_27 and HW_28) based on the maximum apparent temperature (feels like temperature, similar to heat-index and humidex) that is computed using temperature and humidity. We selected these definitions from literature search focusing on studies evaluating association between heat waves and health outcomes.
Reviewer 3 Report
Summary: The authors evaluated the ability of 28 different heat wave definitions to best predict the number of heat-related ED visits across portions of the North Carolina (NC) Coastal and Piedmont regions. Overall, the manuscript is well motivated, well written, and well organized. My concerns are minimal except for one glaring omission that, once remedied, should improve their message and reception by the target audiences. Therefore, I am recommending minor revisions.
Major Concern and Recommendation:
1. Much of the articulated motivation for this sub-regional study is significant climatic heterogeneity present over larger regional studies. Unfortunately, the authors do not show this heterogeneity for their North Carolina study area. A map is needed! The map could show the state of North Carolina divided into (a) counties and (b) the three physiographic regions. Overlain on this information could be the county warning area (CWA) boundaries for the three National Weather Service offices for which heat products are evaluated. This map could be accompanied by a second map showing population density across the state. Collectively, these two maps would provide a nice geographical perspective as to what portions of the state and the main population centers are include in the study.
Minor Concerns and Recommendations:
2. Lines 79-81: Given that the mountain region is ultimately excluded from the study (explained on Lines 148-149), it would be helpful to state this at this stage – when you are also excluding data from 2013.
3. Lines 139-149: Provide more discussion as to how the ED visit data are stratified and aggregated. Is the data provided on the county level (and thus you only pulled ED data for those counties included in the CWAs of the three NWS sites)? Or is the data aggregated differently, such that the temperature data, NWS heat products, and ED data are effectively covering different regions? These details need to be included. Again, the maps suggested above would be helpful.
4. Lines 182-184: It would be helpful to state in a parenthetical which two heat wave definitions had the best fit for each region (e.g., HW_XX).
5. Lines 269-270: Use maps to show the heterogeneity within your study region.
6. Line 274: This statement is not correct. In fact, three NWS WFOs cover the NC Piedmont region (Raleigh, Blacksburg, and Greenville-Spartanburg). Again, maps would be helpful to show which portions of NC are (and are not) covered by your study.
Author Response
Major Concern and Recommendation:
- Much of the articulated motivation for this sub-regional study is significant climatic heterogeneity present over larger regional studies. Unfortunately, the authors do not show this heterogeneity for their North Carolina study area. A map is needed! The map could show the state of North Carolina divided into (a) counties and (b) the three physiographic regions. Overlain on this information could be the county warning area (CWA) boundaries for the three National Weather Service offices for which heat products are evaluated. This map could be accompanied by a second map showing population density across the state. Collectively, these two maps would provide a nice geographical perspective as to what portions of the state and the main population centers are include in the study.
Thank you for your suggestion. As suggested, we added three maps describing physiographic regions, WFOs within physiographic regions and total population as supplement figure 1.
Minor Concerns and Recommendations:
- Lines 79-81: Given that the mountain region is ultimately excluded from the study (explained on Lines 148-149), it would be helpful to state this at this stage – when you are also excluding data from 2013.
Thank you for the recommendation. We included information on excluding mountain region in line 92-93, “Additionally, Mountain region was excluded from the analysis as 50.13% of the data was censored due to low HRI visits.”
- Lines 139-149: Provide more discussion as to how the ED visit data are stratified and aggregated. Is the data provided on the county level (and thus you only pulled ED data for those counties included in the CWAs of the three NWS sites)? Or is the data aggregated differently, such that the temperature data, NWS heat products, and ED data are effectively covering different regions? These details need to be included. Again, the maps suggested above would be helpful.
Thank you, for your suggestion. We added description on line 159-161, “We considered NWS heat wave days among physiographic regions if one or more counties within the physiographic regions had heat warnings/advisories.”
- Lines 182-184: It would be helpful to state in a parenthetical which two heat wave definitions had the best fit for each region (e.g., HW_XX).
Thank you for your comment. We added information in line 209-210, “The statistical model with the lowest AIC value among the 28 HWDs was considered the optimal HWD (Coastal: HW_15 and Piedmont: HW_07).”
- Lines 269-270: Use maps to show the heterogeneity within your study region.
Thank you for your comment. As suggested, we added maps as supplement figure 1.
- Line 274: This statement is not correct. In fact, three NWS WFOs cover the NC Piedmont region (Raleigh, Blacksburg, and Greenville-Spartanburg). Again, maps would be helpful to show which portions of NC are (and are not) covered by your study.
Thank you, for your comment. We provided detailed information on the coverage of WFOs in line: 143-149 and line 328-331. There were heat products (advisories and warnings) issued by Wilmington, Newport, and Raleigh. Among the three WFOs, Wilmington and Newport covered the Coastal region; Raleigh covered Piedmont region.
Round 2
Reviewer 2 Report
Thank you for your appropriate response. I accept your article for press in IJERPH.